# Spatial competition constrains resistance to targeted cancer therapy

Katarina Bacevic[1], Robert Noble [2,5], Ahmed Soffar [1,6], Orchid Wael Ammar[1], Benjamin Boszonyik[1], Susana Prieto[1], Charles Vincent[3], Michael E. Hochberg[2,4], Liliana Krasinska[1] & Daniel Fisher[1]

Adaptive therapy (AT) aims to control tumour burden by maintaining therapy-sensitive cells to exploit their competition with resistant cells. This relies on the assumption that resistant cells have impaired cellular fitness. Here, using a model of resistance to a pharmacological cyclin-dependent kinase inhibitor (CDKi), we show that this assumption is valid when competition between cells is spatially structured. We generate CDKi-resistant cancer cells and find that they have reduced proliferative fitness and stably rewired cell cycle control pathways. Low-dose CDKi outperforms high-dose CDKi in controlling tumour burden and resistance in tumour spheroids, but not in monolayer culture. Mathematical modelling indicates that tumour spatial structure amplifies the fitness penalty of resistant cells, and identifies their relative fitness as a critical determinant of the clinical benefit of AT. Our results justify further investigation of AT with kinase inhibitors.

[1] IGMM, CNRS, University of Montpellier, 34090 Montpellier, France. [2] ISEM, University of Montpellier, 34090 Montpellier, France. [3] IRCM, Inserm, 34090 Montpellier, France. [4] Santa Fe Institute, Santa Fe, NM 87501, USA. [5]Present address: Department of Biosystems Science and Engineering, ETH Zürich, 4058 Basel, Switzerland. [6]Present address: Division of Molecular Biology, Department of Zoology, Faculty of Science Alexandria University, 21526 Alexandria, Egypt. Katarina Bacevic and Robert Noble contributed equally to this work. Michael E. Hochberg, Liliana Krasinska and Daniel Fisher jointly supervised this work. Correspondence and requests for materials should be addressed to D.F. (email: daniel.fisher@igmm.cnrs.fr)

Kinase inhibitors targeting signaling pathways have shown major value in targeted cancer therapies but generally fail due to acquired resistance[1, 2]. Numerous studies have identified activation of alternative signaling pathways as possible resistance mechanisms (e.g., ref. [3]), suggesting that combination therapies directed against multiple pathways would be beneficial. As an alternative strategy, adaptive therapy (AT) is proposed to be advantageous in such settings, and more effective at controlling resistance than conventional maximal tolerated dose (MTD) approaches[4–8]. In AT, therapeutics are used at low-dose, adjusted to maintain tumour burden constant rather than eradicating all tumour cells. This in theory preserves therapy-sensitive cells that will outcompete resistant cells, due to the reduced proliferative fitness of the latter. This assumption has not been validated. Furthermore, whereas previous mathematical modelling[7] indicated that AT should confer a large survival benefit, this model assumed that the relative fitness of resistant cells is proportional to their frequency in the population. As such, the relative fitness of rare resistant cells would approach zero, which is unlikely. Crucially, experimental investigations of AT did not monitor resistance frequency nor measure cell fitness. In mouse xenograft models using cytotoxic chemotherapy, combining one MTD dose followed by lower doses resulted in better long-term tumour control than the MTD treatment alone[4, 6]. Although this result might indeed reflect reduced selection for resistance, alternatively, it may have been due to the higher cumulative drug dose applied. The principles underlying AT thus remain unproven.

To test the assumptions of AT, we developed a new mathematical model of the population dynamics of therapy-sensitive and resistant cells, and an experimental system allowing us to test its predictions. We hypothesised that resistance to inhibitors of cell cycle regulators would likely incur a fitness cost, potentially fulfilling the assumptions of AT and allowing us to test which parameters are critical. We focused on cyclin-dependent kinases (CDKs), which control the cell cycle and whose pathways are universally deregulated in cancer[9]. Small molecule CDK inhibitors (CDKi) have been developed as agents for cancer therapy. Early clinical trials with non-specific CDKi showed promising responses but were hindered by toxicity[10]. In 2015, palbociclib (PD0332991), which targets CDK4 and CDK6, was approved for use in cancer therapy[11, 12]. However, not all cancer cells respond to CDK4/6 inhibition, and loss of RB1 renders cells insensitive[13–16]. Yet probably all cancer cells have active CDK1 and CDK2. CDK1 is essential for cell proliferation[17, 18], whereas CDK2 knockout mice are viable[19, 20] and CDK2 knockdown is tolerated by most cancer cells[21]. Nevertheless, acute pharmacological or peptide-based inhibition of CDK2 strongly inhibits cancer cell proliferation[22–25], CDK2 counteracts Myc-induced cellular senescence[26] and CDK2-knockout mouse cells are resistant to oncogenic transformation[19]. Thus, CDK1 or CDK2 inhibition will likely have therapeutic benefits.

We predicted that resistance to CDK1/CDK2 inhibitors might arise through alteration of cell cycle pathways, reducing proliferative fitness. We therefore generate colorectal cancer cells with acquired resistance to a CDK1/CDK2-selective inhibitor, and identify mechanisms of resistance. These involve stable rewiring of cell cycle pathways, resulting in compromised cellular fitness. Based on competition experiments with different treatment regimes and computer simulations, we find that tumour spatial structure is a critical parameter for AT. Competition for space increases fitness differentials, allowing effective suppression of resistant populations with low-dose treatments.

## Results

**Mathematical modelling of tumour evolution under AT.** To investigate the hypothesis that AT might control tumour growth more effectively than MTD, we first developed a new minimally complex mathematical model of tumour evolutionary dynamics during therapy to capture the fundamental dynamics of AT and MTD. Previous mathematical modelling[7] indicated that AT could confer very large survival benefit, that strongly depended on the fraction of resistant cells in the population (frequency) when treatment begins. However, relative fitness of resistant cells was assumed to be proportional to their frequency (Fig. 1a, solid line), a probable oversimplification of dynamics in situ. The premise underlying AT is that, on average, resistant cells proliferate more slowly when surrounded by sensitive cells than other resistant cells. Yet competition for diffusion-limited resources is generally confined to relatively small neighbourhoods, and a change in frequency below or above certain thresholds should not much affect resistant cell fitness. From these considerations and a geometrical analysis of resistant subclone growth within a three-dimensional tumour (Supplementary Methods; Supplementary Fig. 1), we propose that the relationship between the relative fitness and the frequency of resistant cells can be more realistically represented by a sigmoidal function, with its lower asymptote greater than zero (Fig. 1a, dashed lines). The two asymptotes correspond to the relative fitness of resistant cells when they are either (i) surrounded by drug-sensitive tumour cells that constrain their population growth (lower), or (ii) more abundant and have escaped from competition with drug-sensitive cells (upper). We assume that the transition between fitness levels is relatively abrupt as resistant cells that escape competition with sensitive cells will rapidly expand. Yet our model predicts that the relative benefits of AT are insensitive to the exact frequency of the transition (Supplementary Methods; Supplementary Fig. 1).

We used coupled differential equations to model population dynamics. We obtained approximate analytical solutions to determine how treatment outcomes depend on biological parameters (Supplementary Methods). To confirm their validity and to facilitate comparisons between studies, we also ran numerical simulations for AT and MTD regimes that were examined in previous analysis[7]. In simulations of MTD, therapy was applied as a constant-dose bolus at regular intervals. For AT, we began with half the MTD dose and adjusted it by 20% if the total population size had increased or decreased since the previous treatment. The AT dose was not allowed to exceed that of MTD. The cell number at treatment onset was set at $10^9$, approximating the cell population in a typical human tumour at first detection (1 cm$^3$). We compared the predicted survival time (defined as the time taken for the tumour cell population to reach $10^{12}$), and progression-free survival time (defined as the time taken for the tumour to regain its pre-therapy size) between AT and MTD regimes.

With the previously-described linear fitness function[7], AT can stall tumour growth indefinitely, provided resistance is sufficiently rare at the start of treatment and resistant cells maintain some sensitivity (Fig. 1b; Fig. 1c, black solid line). Even if some cells are 100% resistant, the benefit of AT, relative to MTD, is unbounded and increases rapidly with decreasing initial resistance frequency (Fig. 1c, black dashed line).

Conversely, with our more plausible sigmoidal fitness function, the benefit of AT is predicted to be more modest (Fig. 1b). The relative benefit of AT for progression-free survival (Fig. 1c) or overall survival (Supplementary Fig. 2a), is limited by an upper bound that is independent of the initial resistance frequency but varies with the maximum growth rate of resistant cells (Fig. 1d) and the degree of resistance (Supplementary Fig. 2b). For realistic

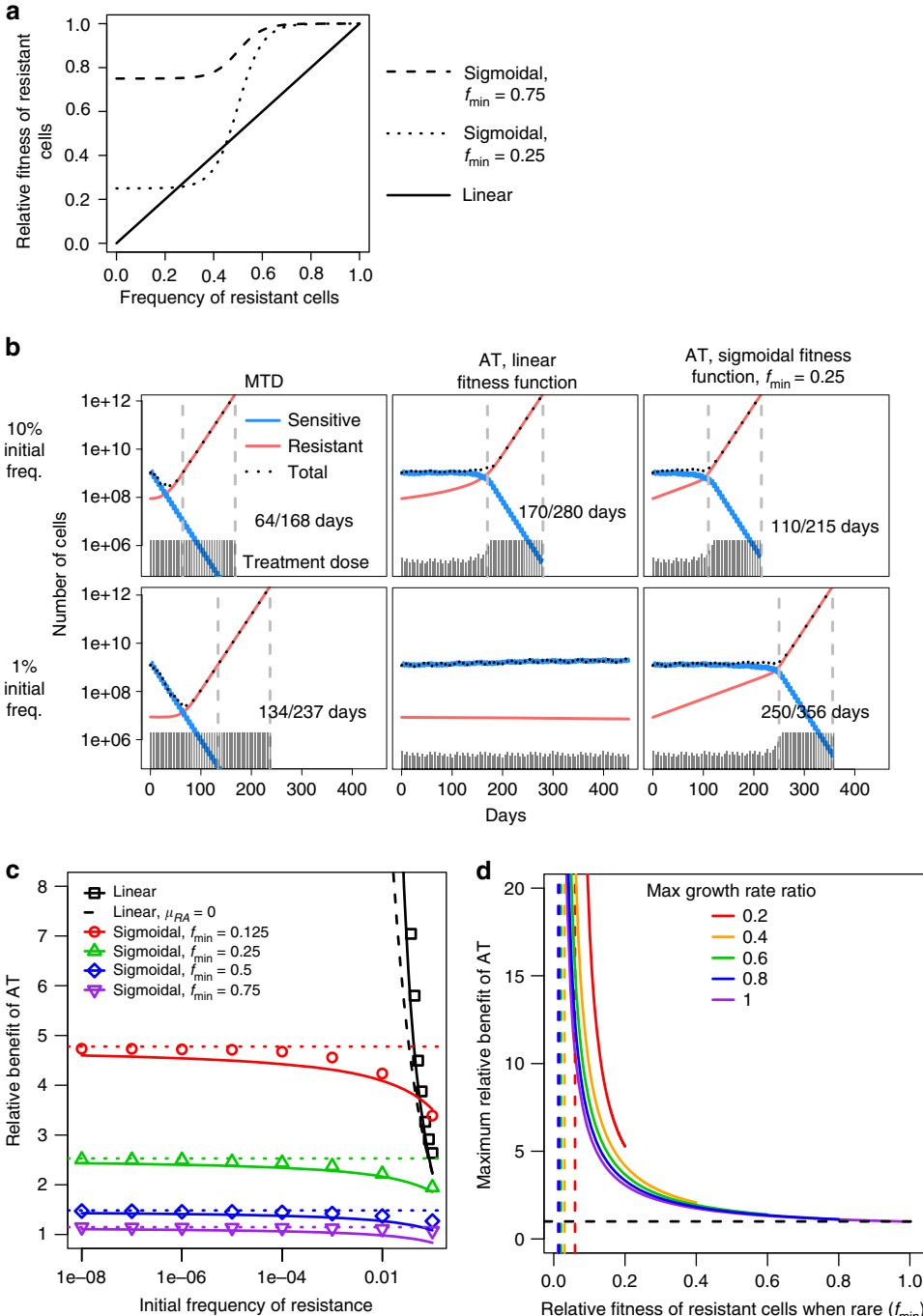

**Fig. 1** Mathematical modelling of tumour evolutionary dynamics. **a** In mathematical modelling of cancer treatment outcomes, the function $f$ describing the relationship between resistant cell relative fitness and frequency may be assumed to be linear (solid curve) or sigmoidal. Two example relationships are shown for the relative fitness of resistant cells when rare, $f_{min} = 0.25$ (dotted curve) and $f_{min} = 0.75$ (dashed curve). **b** Numerical results of a mathematical model for different therapy regimes, varied initial frequency of resistant cells, and varied function $f$. Population sizes are shown for sensitive cells (blue), resistant cells (red), and all cells (black). Days of progression-free survival (first vertical dashed line) and overall survival (second vertical dashed line) are shown. Grey vertical bars show the therapy dose. MTD = maximum tolerated dose; AT = adaptive therapy. **c** Mathematical model predictions for the progression-free survival benefit of adaptive therapy (relative to maximum tolerated dose therapy) vs. initial frequency of resistant cells. Symbols represent numerical results; lines are approximate analytical solutions; dotted lines are the upper bounds of the approximate analytical solutions. Outcomes are shown for five models assuming different functions $f$. The first model (black solid curve and square points) assumes a linear function, and assumes that the therapy slightly increases the mortality rate of resistant cells, $\mu_R$. An analytical approximation is also shown for the case $\mu_R = 0$ (dashed black line). Other models assume that $f$ is sigmoidal. **d** Maximum relative survival benefit of adaptive therapy vs. $f_{min}$, assuming a sigmoidal function $f$. Curves are shown for different values of $\lambda_R/\lambda_W$, which is the maximum growth rate of resistant cells, relative to the growth rate of sensitive cells. The vertical asymptotes are at $\mu_{RA}/\lambda_R$ and the horizontal asymptote is at 1. Parameter values are taken from a previous study[7] to facilitate comparison. Unless specified otherwise, $\lambda_W = \lambda_R = \log(2)/10$, $IC50_W = 1$, $IC50_R = 100$, $\rho_{MTD} = 1$, $\theta = 5$ days, $N_0 = 10^9$, $k = 20$, and $c = 0.5$

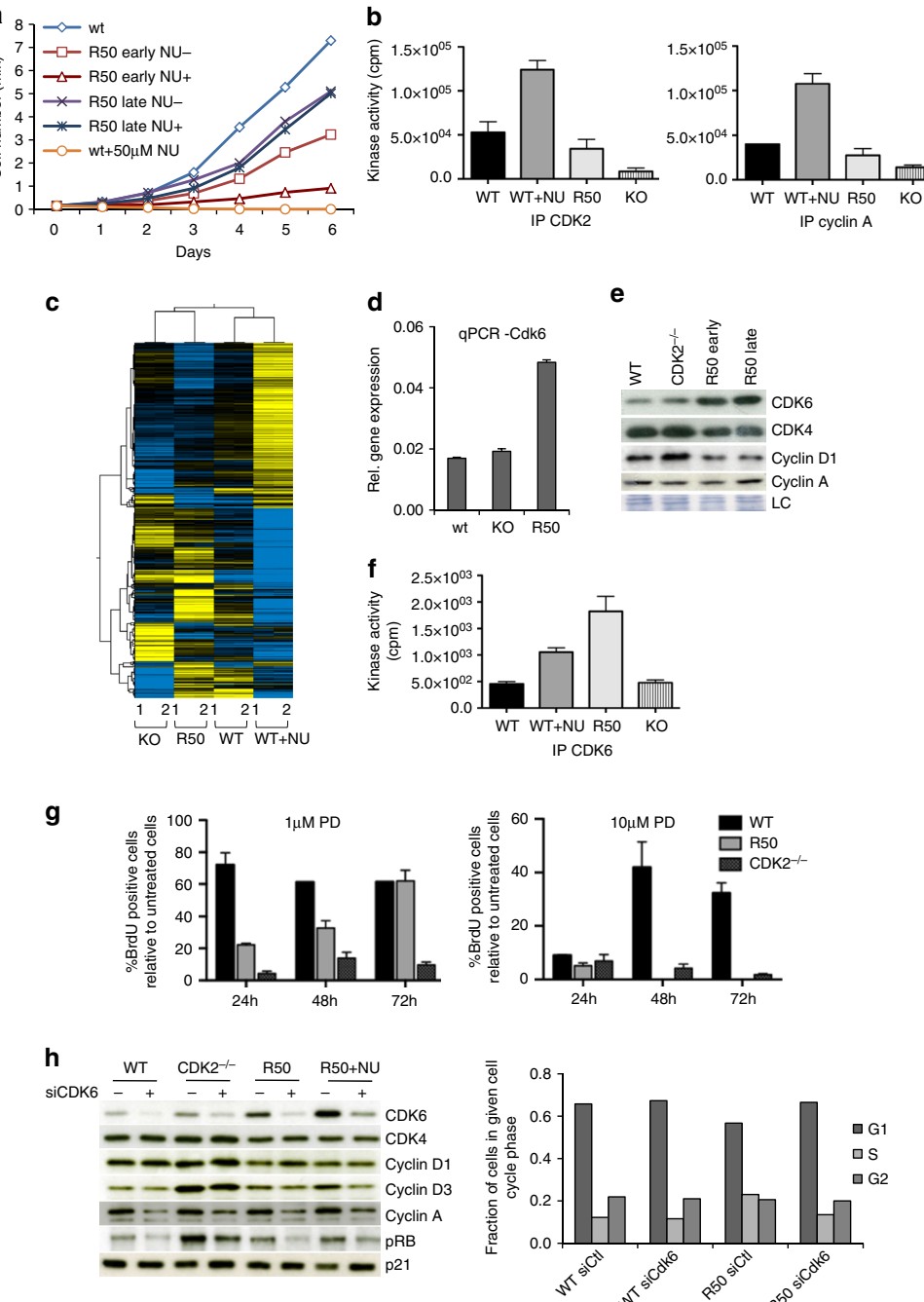

**Fig. 2** Cell cycle rewiring in CDKi-resistant cells. **a** HCT116 WT and R50 early and late cells were grown in the absence or presence of 50 μM NU6102 and the cell number was evaluated every day for 6 days. Representative of three independent experiments. **b** Kinase activity assayed in vitro on CDK2 and cyclin A immunoprecipitates from WT, WT cells treated for 24 h with 20 μM NU6102, R50 and CDK2$^{-/-}$ (KO) cells (mean±SD of 2 independent experiments performed in triplicates). **c** Heat-map of 2300 genes differentially expressed between WT, CDK2$^{-/-}$ (KO), R50, and WT cells treated with 20 μM NU6102 for 24 h; analysis was performed in duplicates. Upregulated genes are indicated in yellow, downregulated in blue. **d** qRT–PCR analysis of CDK6 expression in WT, CDK2$^{-/-}$ (KO) and R50 cells (mean±SD of three replicates). **e** Western blot analysis of the proteins indicated in WT, CDK2$^{-/-}$ and R50 early and late cells. **f** Kinase activity assayed in vitro on CDK6 immunoprecipitates from WT, WT cells treated for 24 h with 20 μM NU6102, R50 and CDK2$^{-/-}$ (KO) cells (mean ±SD of two independent experiments performed in triplicates). **g** WT, R50 and CDK2$^{-/-}$ cells were grown in the absence or presence of palbociclib (1 μM or 10 μM; PD0332991, PD) for 24 h, 48 h and 72 h, pulsed for 15 min with BrdU, and analysed for BrdU incorporation relative to untreated cells (mean±SD of a representative experiment performed in triplicates). **h** CDK6 was downregulated with siRNA in WT, CDK2$^{-/-}$, R50 and WT cells treated with 20 μM NU6102 for 24 h. After 24 h, the expression of indicated proteins (left) and cell cycle distribution (right) were analysed

parameter values, the upper bound can be approximated as

$$\frac{1}{f_{min}} - \frac{\lambda_W}{\mu_W}\left(\frac{1}{f_{min}} - 1\right), \qquad (1)$$

where $f_{min}$ is the relative fitness of resistant cells when they are rare; $\lambda_W$ is the division rate of sensitive cells without treatment; $\mu_W$ is the death rate of sensitive cells under MTD (Supplementary Methods). The maximum benefit of AT is thus strongly dependent on $f_{min}$, and increases as $f_{min}$ decreases.

The same pattern holds for less frequent dosing (Supplementary Fig. 2c) and for sigmoidal rather than exponential growth curves (Supplementary Fig. 2d). In summary, this model predicts that AT will only have limited survival benefit over MTD unless the fitness differential between therapy-sensitive and resistant cells is large (at least a factor of two).

**Cell cycle rewiring in CDKi-resistant cells.** To test how the results of mathematical modeling compare with experimental data, we first generated CDK1/CDK2 inhibitor-resistant (ir) cells and analysed their fitness. We chose the CDKi NU6102 since we found that it is CDK2-selective at low doses and we could generate NU6102-resistant alleles of CDK2 by engineering combinatorial mutations in the kinase domain[27]. We confirmed that NU6102 had similar growth-inhibitory effects in several cancer cell lines of different origins and in non-transformed fibroblasts (Supplementary Fig. 3a). In all cell lines, growth was arrested at 20 µM, whereas 50 µM caused major cell death. We chose the colorectal cancer cell line HCT-116 which can rapidly evolve resistance to cell cycle kinase inhibitors[28]. We could also compare CDK inhibition by NU6102 with CDK2 gene deletion (CDK2$^{-/-}$)[29]. We found that CDK2$^{-/-}$ cells were less sensitive to NU6102 than wild-type (WT) cells (Supplementary Fig. 3b), confirming that loss of CDK2 confers partial resistance to a CDK1/CDK2-selective inhibitor.

We stably expressed eGFP in the parental HCT-116 cells, to distinguish them from resistant cells in mixed cultures. To obtain NU6102-resistant cells, we applied either escalating concentrations of NU6102 (up to 10, 20 or 50 µM), or maintained the same concentrations from the start. We obtained NU6102-resistant colonies, which we designated R10, R20 or R50 (e.g., R50 is resistant to 50 µM NU6102), but no line was totally impervious to the inhibitor (Supplementary Fig. 3c). Without NU6102, populations resistant to higher concentrations grew more slowly, indicating compromised fitness (Supplementary Fig. 3c). We next profiled the expression of cell cycle regulators in control, CDK2$^{-/-}$ and resistant cells (Supplementary Fig. 3d). In resistant and control cells, CDK2, CDK1, cyclin A2, cyclin E1, cyclin B1, cyclin D1, p21 and the CDK2 substrates RB and CDC6 were expressed at similar levels, while CDK6 was slightly increased in R20 cells. Cyclin A2, cyclin B1, CDK6, CDC6 and RB were strongly reduced in R50 cells, in accordance with their poor proliferation in the presence of inhibitor. However, after serial passaging in 50 µM NU6102, R50 cell proliferation became totally refractory to the inhibitor (Fig. 2a, compare R50-early and -late). Thus, resistance can evolve. Without inhibitor, R50-early and R50-late cells both proliferated less well than control cells, indicating that resistance indeed incurred a fitness penalty, in agreement with our original prediction. We tested the fraction of cells synthesising their DNA by analysing 5-ethynyl deoxyuridine (EdU) incorporation after either pulse-labelling or 24-hour exposure. This revealed that all R50 and WT cells replicated within a 24-h period, but 20% fewer R50 cells were in S-phase at a given time, indicating an altered cell cycle distribution (Supplementary Fig. 3e). R50 did not have increased apoptosis, as determined by 7-aminoactinomycin D staining and western

blotting for cleaved caspase-3 (Supplementary Fig. 3f). To test whether resistance was reversible, we withdrew the inhibitor for 2 or 6 months ("drug holidays"). Re-exposure to NU6102 did not affect the cell growth, demonstrating that resistance was irreversible (Supplementary Fig. 3g).

We next tested the sensitivity of R50 to other CDKi, and found that they were more sensitive than controls to another tri-substituted purine, purvalanol A[30] (Supplementary Fig. 4a), that also strongly inhibits CDK1 and CDK2[27]. This suggests that cells were not generally drug-resistant and that a specific alteration in CDK pathways contributed to NU6102 resistance. We cloned and sequenced the CDK2 gene from R50 cells and found no mutations. We then measured CDK2 kinase activity by assaying phosphorylation of histone H1 on immunoprecipitates of CDK2 or cyclin A from R50 cells, CDK2$^{-/-}$ cells, WT cells, and WT cells treated with 20 µM NU6102 (Fig. 2b). As expected, only background CDK2 activity was detectable in CDK2$^{-/-}$ cells, though cyclin A-associated activity (which includes CDK1) was less affected. WT cells treated with 20 µM NU6102 had higher maximal CDK2 activity than control cells, an expected result of their arrest in G2/M. Despite having similar levels of cyclin A-CDK2 complexes to WT (Supplementary Fig. 4b), CDK2 activity from R50 cells was reduced (Fig. 2b), perhaps contributing to their NU6102 resistance. To investigate CDK2 activity directly in cells, we used a recently-developed sensor, DHB-Venus[31]. This probe translocates from the nucleus to the cytoplasm upon phosphorylation by CDK2. However, the probe behaved similarly in CDK2$^{-/-}$, R50 and WT cells (Supplementary Fig. 4c), indicating that another kinase can substitute for CDK2 in G1/S. Altogether, these results suggest that while CDK2 activity was reduced in R50, overall CDK activity was comparable, implying rewiring of CDK pathways.

Upregulating Ras signaling pathways and cyclin E expression is involved in resistance to CDKi in HEY ovarian cancer cells[16]. Cyclin E expression was normal in R50 cells (Supplementary Fig. 3d). We probed the activity of major cell signaling pathways by protein phosphorylation array analysis, but found no major alterations between WT cells, WT cells treated with 20 µM NU6102, R50 cells or CDK2$^{-/-}$ cells (Supplementary Fig. 4d). This further highlights the CDK-specificity of NU6102 and is consistent with specific resistance mechanisms. We next investigated gene expression genome-wide by microarray analysis. R50 cells clustered with CDK2$^{-/-}$ cells, and the effects of loss of CDK2 gene or activity (NU6102 resistance) on the transcriptome were distinct from, and generally opposite to, the effects of CDK inhibition (Fig. 2c and Supplementary Data 1). Gene ontology analysis showed that the most altered pathways in R50 cells compared to WT cells ± NU6102 included pathways in cancer, the cell cycle, and RNA transport (Supplementary Fig. 4e). We confirmed by qRT-PCR that CDK6 was highly upregulated in R50 cells (but not CDK2$^{-/-}$ cells) (Fig. 2d). While CDK4 levels were slightly reduced, both CDK6 protein and kinase activity were strongly increased in proliferating R50 cells (Fig. 2e, f). As determined by western blotting immunoprecipitated cyclin D1 and cyclin D3 from R50 cells, CDK6 showed no cyclin D specificity, whereas CDK4 preferentially complexed with cyclin D1 (Supplementary Fig. 4f). These results suggest that upregulation of CDK6 is involved in resistance to CDK2 inhibition. Determination of CDK2, CDK4 and CDK6 enzyme kinetic parameters in vitro (Methods and Supplementary Fig. 4g) showed that CDK6 affinities for ATP and NU6102 were intermediate between those of CDK2 and CDK4. The $K_m$(ATP)/$K_i$(NU6102) ratio, which reflects inhibitor sensitivity, was 2-3-fold lower for CDK6-cyclin D1 than for CDK2-cyclin A2. R50 and CDK2$^{-/-}$ cells were more sensitive than control cells to the CDK4/CDK6 inhibitor PD0332991 (Fig. 2g). CDK6 knockdown by siRNA

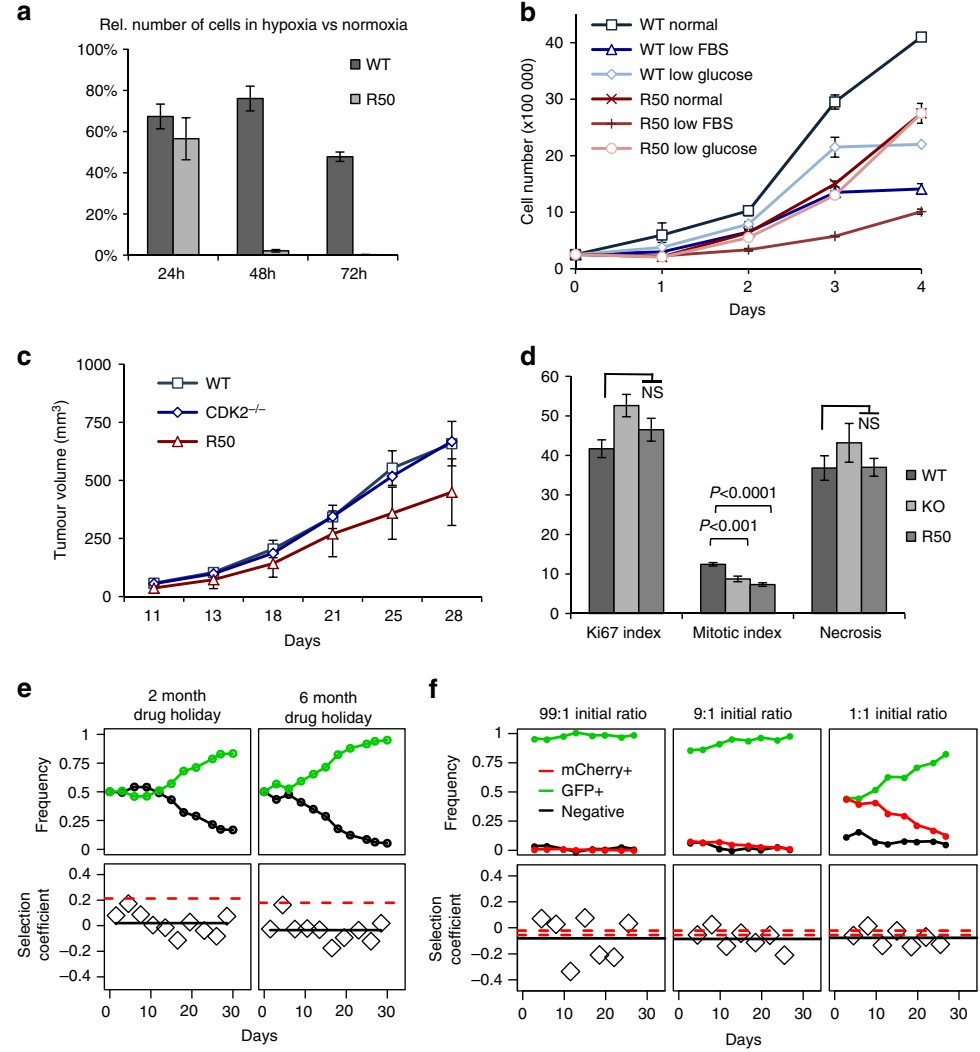

**Fig. 3** CDKi-resistant cells have lower proliferative fitness in vitro and in vivo. **a** WT and R50 cells were grown in 1% O₂. Percentage of viable cells (relative to cells cultured in 21% O₂) was quantified after 24 h, 48 h and 72 h (mean±SD of two independent experiments). **b** WT or R50 cells were grown in low serum (1%) or low glucose (1 g/L) and their number was analysed every 24 h (mean±SEM of 2 independent experiments). **c** Nude mice were injected subcutaneously with WT, CKD2⁻/⁻, or R50 cells, and tumour volume was measured at 3-day intervals ($n = 8$ mice per condition; mean±SD). **d** Immunohistochemistry analysis of mitotic index, Ki67 expression and necrosis (Caspase 3a) in xenograft tumour samples from **c** (represented as % of cells, mean±SEM; t-test, ns, not significant). **e**, **f** Frequency dynamics (top row) and selection coefficients for competition assays, compared to predictions from growth rates (bottom row). In the bottom row, each point corresponds to a selection coefficient calculated from a competition assay (i.e., a period between consecutive points in the top row). Solid lines are means. Red dashed lines indicate predictions based on the growth rates of each cell type in isolation (Supplementary Methods). A single prediction is shown whenever growth curves were measured at the same time as competitions were conducted; otherwise pairs of lines show maximum and minimum predictions based on non-contemporaneous growth curves. Results are shown for competitions between GFP+CDKi-sensitive cells and drug-holiday GFP- CDKi-resistant (R50) cells (**e**), and for competitions between GFP+CDKi-sensitive cells and mCherry+R50 cells with different initial ratios (**f**)

reduced cell proliferation similarly in both WT and R50 cells, as reflected by a decrease in cyclin A and phospho-RB levels (Fig. 2h). However, in R50, the CDK6 level remaining after siRNA was comparable to that of untreated WT cells. Thus, resistance to a CDK2-selective inhibitor led to increased dependency on CDK6, suggesting that CDK2 and CDK6 can perform similar cellular functions in HCT-116 cells.

Next, we tested the physiological relevance for tumourigenesis of this rewiring of CDK pathways. In vivo, tumour formation will depend on sensitivity to limiting oxygen and nutrients. We therefore compared growth curves of GFP + and R50 under conditions of low nutrients or hypoxia. R50 cells were markedly sensitive to culture in 1% oxygen (Fig. 3a), and low serum, but were less sensitive to low glucose (Fig. 3b), indicating possible

effects of CDK rewiring on metabolism. We next subcutaneously injected parental (WT), R50 and CDK2⁻/⁻ cells in nude mice. All three cell lines could form tumours in mice, but R50 grew markedly more slowly than WT and CDK2⁻/⁻ tumours (Fig. 3c). Lower mitotic index indicated impaired cell proliferation (Fig. 3d). Thus, whereas CDK2 gene deletion did not affect tumour cell growth, R50 cells have lower proliferative fitness both in vitro and in vivo.

These results confirmed our initial hypothesis and suggested that, when mixed together in the absence of inhibitor, resistant cells should be outcompeted by sensitive cells. We adjusted results to compensate for loss of eGFP expression (as quantified in control experiments). When seeded at a 1:1 ratio, eGFP-expressing cells were outcompeted by eGFP-negative control

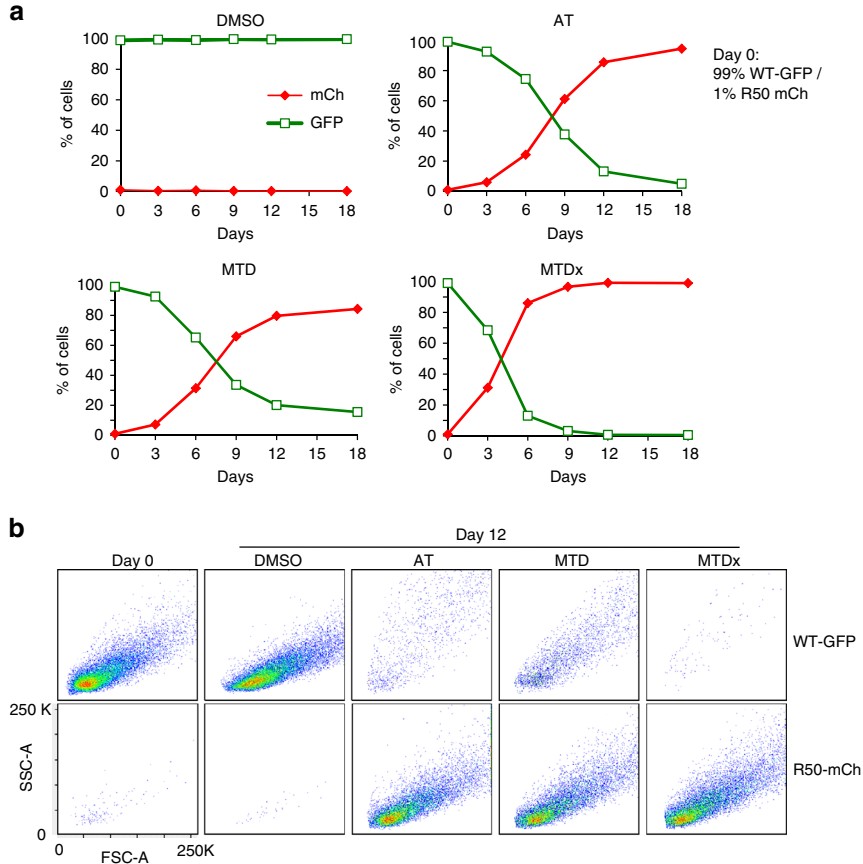

**Fig. 4** AT does not outperform MTD in limiting tumour growth in a monolayer culture. **a** Sensitive GFP+ and R50 mCherry+ cells were plated in monolayer at initial 99:1 ratio, in the presence of either DMSO as control or NU6102: AT condition (15 µM initial drug concentration, ±20% at 3-day intervals to maintain stable 70–80% confluence), MTD (50 µM for 24 h, with 48 h no drug) or MTDx (continuous 50 µM). Every 3 days, the proportion of GFP+ and mCh+ cells was determined by flow cytometry. **b** Flow cytometry analysis (forward scatter area, FSC-A, which increases with cell size, and side scatter area, SSC-A) of cell size of WT-GFP+ and R50-mCh+ cells from the experiment in **a**, at day 0 and day 12

cells, indicating impairment of growth due to eGFP expression (Supplementary Fig. 5a). We then competed R50 cells with eGFP-expressing control cells in equal initial proportions, in the absence of inhibitor. Although R50 cells initially grew faster, they were later outcompeted by control cells (Supplementary Fig. 5a). This indicates that resistant cells are significantly less fit than control cells in long-term co-culture. Their initial higher competitiveness suggested a possible addiction to inhibitor, so to test this we competed eGFP control cells with R50 cells that had been grown without inhibitor for 2 or 6 months ("drug holiday"). Indeed, R50 drug-holiday cells were even less fit and had no initial growth advantage (Fig. 3e). For each competition we estimated the selection coefficient, which measures the fitness difference between the two cell lines and corresponds to the difference between the growth rates of the two cell lines grown separately. R50 cells growing with eGFP cells had a selection coefficient similar to that predicted from the difference between the monoculture growth rates (Supplementary Fig. 5a). In contrast, the selection coefficient of drug-holiday R50 cells was substantially lower than expected (Fig. 3e), indicating further adaptation that reduces their competitiveness, i.e., drug addiction.

To offset the fitness differential caused by expression of eGFP and to allow direct quantitation of the resistant and sensitive populations, we generated R50 cells stably expressing mCherry. We grew each cell line either alone or in mixed cultures at different initial ratios, and quantified populations over a one-month period by flow cytometry. R50-mCherry cells (R50-mCh) were outcompeted by WT-eGFP cells, and the selection

coefficient was approximately equal to the difference in mono-culture growth rates (Fig. 3f, Supplementary Fig. 5b).

**Effective AT requires spatially structured growth**. Our mathematical model predicts that AT performs substantially better than MTD only if resistant cells are relatively much less fit (>2-fold) than sensitive cells when resistance is rare. Our CDKi R50 cells had a relative fitness of 90–95%, regardless of their frequency, suggesting that AT would not be advantageous over MTD in slowing the growth of resistant cells in monolayer culture.

To test this prediction, we first mimicked AT and MTD therapy by treating mixed monolayer cultures of R50-mCh and control GFP+ cells (seeded at 1:99 initial ratio) with either 50 µM NU6102 to kill sensitive cells (MTD, with or without [MTDx] a 2-day break between drug treatments), or an initial concentration of 15 µM NU6102, i.e., just below the concentration required to maintain a stable population (AT). We modified the concentration of NU6102 in AT by ± 20% at 3-day intervals to maintain a constant cell density. We determined the proportions of R50-mCh and GFP + at different time points by flow cytometry. As expected, MTD simply eliminated GFP+ cells while R50-mCh cells grew freely (Fig. 4a; Supplementary Fig. 6a). Although AT arrested growth of GFP+ cells (while not proliferating, they increased in size, reminiscent of senescent cells; Fig. 4b; Supplementary Fig. 6b), it had negligible effect on R50-mCh cell growth. We tested different concentrations of NU6102 for AT treatment, but under no condition did GFP+ cells strongly

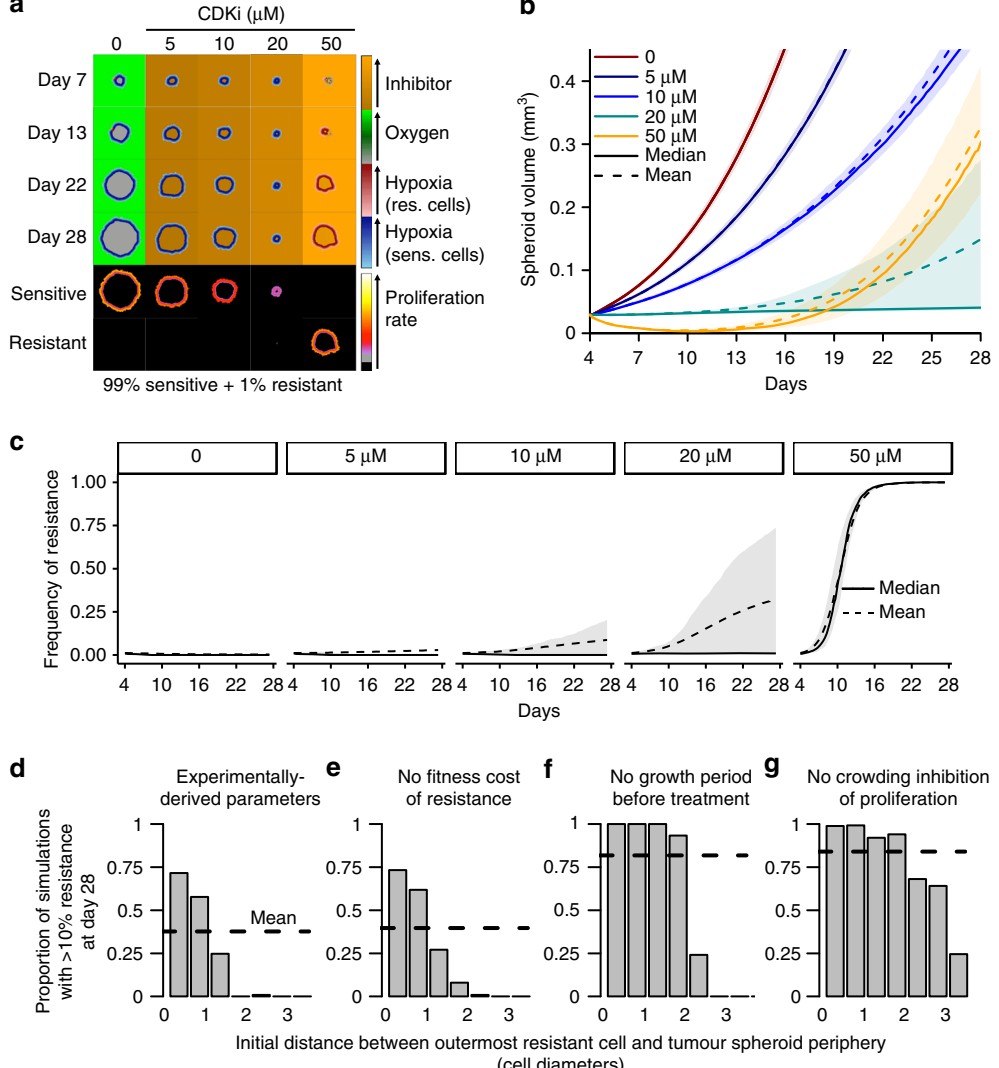

**Fig. 5** Spatial structure is a critical determinant of AT efficiency. **a** Time series of tumour spheroid cross-sections in a spatial computational model. In the upper four rows, cells are coloured according to their type (blue = sensitive, red = resistant) and shaded by local oxygen concentration (darker = more hypoxic); the medium surrounding the tumour spheroid and the necrotic core are coloured according to CDKi concentration (pale orange = high, dark orange = low) or, in the absence of CDKi, are coloured according to oxygen concentration (pale green = high, grey = low). In the lower two rows, cells are coloured according to their proliferation rate (grey = zero, red = low, yellow = high). Typical outcomes were chosen as those in which the final frequency of resistance most closely matched the median. See text and Supplementary Methods for parameter values and assumptions. **b** Tumour spheroid growth curves for different treatment regimens in the computational model. Medians (solid curves), means (dashed curves) and interquartile ranges (shaded) are shown for 1,000 stochastic simulations. **c** Frequency of resistance over time in the computational model. Medians (solid curves), means (dashed curves) and interquartile ranges (shaded) are shown for 1000 stochastic simulations. **d–g** Proportion of simulations in which the frequency of resistance exceeded 10% after 24 days of treatment with 20 μM CDKi, vs. the initial distance between the outermost resistant cell and the tumour spheroid periphery. Results are shown for 1000 stochastic simulations each of the default model (**d**); a model without a fitness cost of resistance (**e**); a model with no growth period prior to CDKi treatment (**f**); and a model in which cell crowding does not prohibit proliferation (**g**)

compete with, and hinder, growth of resistant cells, thus confirming the predictions of our mathematical model.

In the above experiments, cells were grown in monolayers and it was essential to avoid confluency to allow continued measurement of cell growth and adjustment of AT doses. Competition for space was therefore minimal, and the relative lower fitness of resistant cells and their dependence on higher oxygen and growth factors might not have been fully exploited.

We reasoned that fitness differentials might be further accentuated by spatial structure. To test this hypothesis, we created a computational model (Supplementary Methods, Supplementary Fig. 7a, b). We devised a more sophisticated model to investigate how fitness differentials can arise from

competition for space and oxygen in tumour spheroids, a system that recapitulates the spatial cellular interactions and the resource gradients found in solid tumours. The model comprises subpopulations of sensitive and resistant cells that divide and die stochastically at rates dependent on local concentrations of oxygen and CDKi, which diffuse from the surrounding medium. To account for crowding effects, cells are able to divide only if there is sufficient nearby space. In the absence of CDKi, these factors decrease cellular fitness along a gradient from the tumour spheroid periphery toward the core. Based on our results, we assumed that, without CDKi, the proliferation rate of resistant cells is 90% of that of sensitive cells. CDKi effects on cell proliferation and death rates were also derived from our

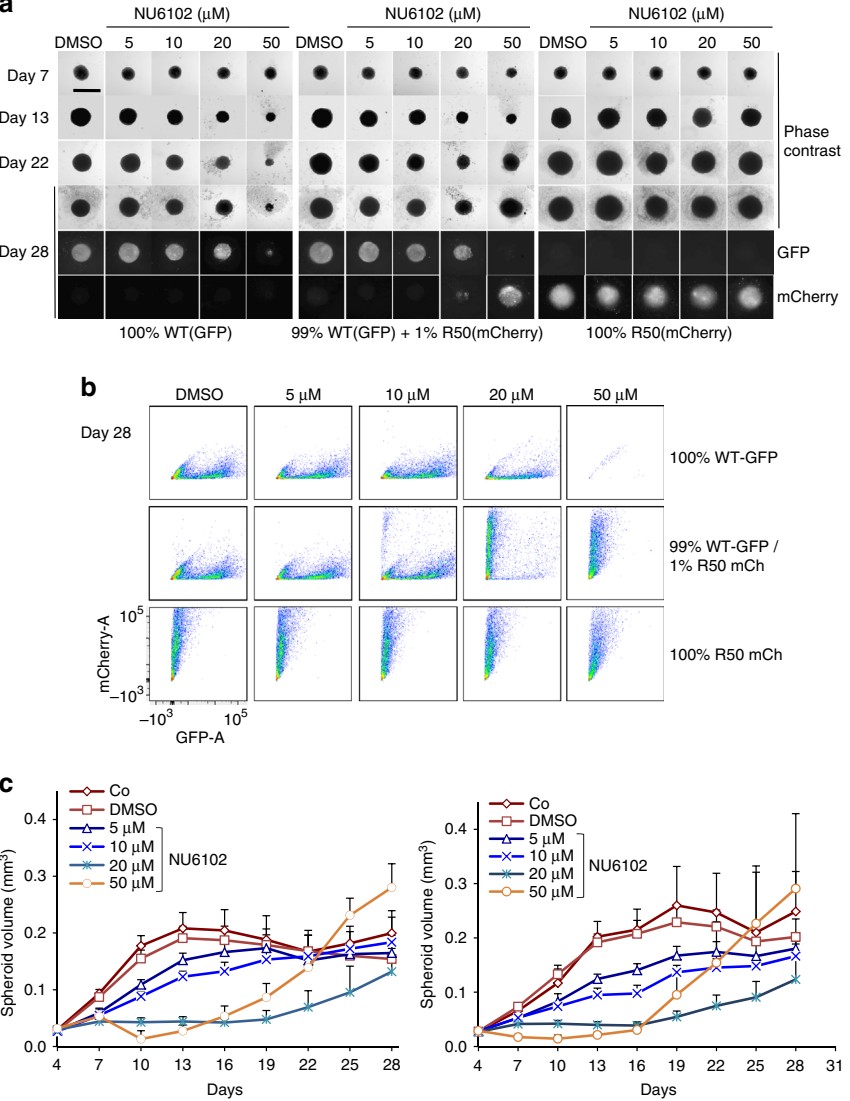

**Fig. 6** Effective AT requires spatially structured tumour growth. **a–c** Tumour spheroids were initiated with WT-GFP+ and R50-mCh+, at different ratios (100% GFP+, 99%GFP+ / 1%mCh+, 100%mCh+), and when established, from day 4, grown either in the presence of DMSO or NU6102 at indicated concentrations. **a** Representative phase contrast and immunofluorescence images of spheroids at indicated time points. Bar, 1mm. **b** Flow cytometry analysis of GFP + and mCh+ cell content of spheroids at day 28 (cells of 4–5 spheroids per condition were dissociated and analysed). **c** Spheroid volume ($mm^3$) at 3-day intervals ($n \geq 5$, mean$\pm$SD). Two independent experiments are presented

experimental data, whereas parameter values related to oxygen were obtained from the literature (Supplementary Methods; Supplementary Fig. 7c). To test the general validity of the model, we disregarded the higher sensitivity of R50 to hypoxia. The code used for tumour spheroid simulations is available under a permissive free software license[32]. We started the model with 1% resistant cells randomly distributed within the tumour spheroid and allowed an initial competition period before introducing CDKi, during which the spheroid increased in volume from 0.07 $mm^3$ to 0.3 $mm^3$. During treatment for 24 days at 20 µM CDKi, the median increase of the tumour spheroid volume was 38%, whereas at 50 µM it was 930% (Fig. 5a, b; Supplementary Movies 1–5). The median frequency of resistance was 0.8% after 24 days at 20 µM but reached 100% over the same period at 50 µM CDKi (Fig. 5c). At 20 µM CDKi, resistant cells were much more likely to increase in frequency if they were initially located close to the spheroid periphery (Fig. 5d), where abundant space and oxygen confer high relative fitness.

To further investigate the effects of spatial structure, we varied the model's parameter values and the time of treatment initiation. Removing the fitness cost of resistance had a relatively small effect, compared to the effect of the initial location of resistant cells (Fig. 5e). In contrast, removing the competition during the initial growth period up to the spheroid volume of 0.3 $mm^3$ when the treatment was started, and with a random spatial distribution of resistant cells, resulted in much faster population growth (median 950% increase in volume over 24 days) and higher final resistance frequency (median 74% after 24 days) (Fig. 5f). This is because resistant cells located close to the periphery have higher relative fitness. Finally, when we removed crowding inhibition of proliferation, so that even cells far from the periphery were able to proliferate (at rates dependent on local oxygen and CDKi concentrations), tumour spheroids grew faster at the 20 µM CDKi dose than at 50 µM (Supplementary Fig. 7d), and the frequency of resistance after 24 days typically reached 100% (Fig. 5g). Thus, in summary, lower doses better control tumour

growth and resistance than high doses, but only when resistant cells are spatially constrained by sensitive cells.

To test the model experimentally, we set up competition experiments in 3D by growing tumour spheroids. We mixed R50-mCh and control GFP + cells at a 1:99 ratio and maintained them in various concentrations of NU6102. Without CDKi, tumour spheroids grew exponentially to a diameter of around 400 μm, and R50-mCh cells were undetectable, confirming their reduced fitness (Fig. 6a, b; Supplementary Fig. 8). Since we could not wash out the inhibitor without disrupting or losing spheroids, we mimicked AT and MTD using constant low, intermediate or high concentrations of the drug. There was good qualitative agreement between the results of the computational model and those of the experimental system (Figs. 5a, b, and 6a, c; Supplementary Movies 1–5), except that the simulated tumour spheroids were able to grow beyond the maximum size attainable in vitro. At 50 μM NU6102 (MTD), growth of spheroids was initially arrested, but then, as expected, spheroids rapidly grew back and became entirely composed of R50-mCh cells (Fig. 6a–c; Supplementary Fig. 8). Importantly, at lower NU6102 concentrations, overall tumour growth was slightly (10 μM) or strongly (20 μM) inhibited, and spheroids were overwhelmingly composed of GFP + cells (Fig. 6c; Supplementary Fig. 8). Thus, low concentrations of the drug were more effective than high concentrations in restraining both tumour growth and proliferation of resistant cells. This result provides strong support for the AT hypothesis as it shows that by reducing the selective pressure imposed by therapy to allow tumour maintenance rather than eradication, fitness differentials can be exploited to limit emergence of resistance. That this result could be achieved only in tumour spheroids, and not in monolayer culture, supports the conclusion drawn from our computational modelling: spatial structure that reduces the average relative fitness of resistant cells is a critical factor for effective AT.

## Discussion

Conventional cytotoxic chemotherapies generate unsustainable DNA damage in highly proliferative tissues, killing cancer cells that are sensitised to apoptosis. These regimes are usually based on the MTD, which by definition engenders severe side effects, necessitating treatment-free recovery periods. The discontinuous nature of such treatments may well be a factor in disease relapse and resistance, which is almost invariably encountered in the clinic. Furthermore, the DNA damage resulting from disrupted S-phase or mitosis might contribute indirectly to resistance mechanisms by increasing mutation rates. Targeted therapies are more specific, and include inhibitors of protein kinases that are deregulated in certain cancers. Kinase inhibitors are increasingly used in cancer treatment, but also consistently confront resistance[2]. To limit development of resistance, which emerges as a result of selective pressure of treatment, alternative approaches, such as metronomic chemotherapy and adaptive therapy, have been proposed.

In their seminal description of the AT hypothesis, Gatenby et al. stated[4], "The goal of adaptive therapy is to enforce a stable tumour burden by permitting a significant population of chemosensitive cells to survive so that they, in turn, suppress proliferation of the less fit but chemoresistant subpopulations." The questions we set out to answer were: (1) Does resistance to a candidate AT drug truly incur a cost? (2) Can sensitive cells indeed suppress proliferation of resistant cells? (3) Which parameters are decisive for AT to be effective?

Protocols resembling AT, using human breast and ovarian cancer cell lines, were previously tested in immunodeficient mice using conventional chemotherapeutics[4, 6], and resulted in long-term stabilisation of tumour burden. However, over the course of the experiment, more of the drug was given in "AT" than in the "MTD" arm, and the cumulative dose will likely directly affect the outcome. Furthermore, the treatments used might further have indirect impact through inhibition of proliferation of endothelial cells responsible for angiogenesis, as previously seen in metronomic chemotherapy[33, 34]. MTD-based therapies may work less well in this regard since they cannot be tolerated indefinitely, unlike the lower doses of metronomic or adaptive regimes. In any case, whether the benefit of AT indeed depends on increased overall dose directly affecting tumour cells, competition between therapy-sensitive and therapy-resistant cells[5], or, like in metronomic chemotherapy, effects on angiogenesis, is not known.

To maximize the chances of resistance reducing cell fitness, we chose CDK inhibitors, an emerging class of cancer drugs. We characterised resistant cells in terms of both biochemical mechanisms and fitness in vitro and in vivo. Our results provide strong support for the adaptive therapy hypothesis. We show that tumour cell-intrinsic resistance mechanisms can reduce fitness, and this difference in fitness is amplified by spatially structured tumour growth to a point where lower drug doses are better than higher doses at controlling tumour burden and resistance. This fitness penalty was additionally context-dependent as resistant cells were sensitised to hypoxia and low serum. Nutrients and oxygen become limiting in the interior of tumour spheroids and even more so in vivo. Context-dependent fitness penalties might be generally true for resistance to targeted therapies, as a recent study found that resistance to the EGFR kinase inhibitor erlotinib in lung cancer cells incurred a fitness penalty that varied according to nutrient and oxygen availability[35]. However, modelling shows that even without any context specificity, the combination of spatial confinement and the inherent fitness penalty of resistance allows effective competition between sensitive and resistant populations. In this context, AT is advantageous.

That our results were obtained in vitro (and reproduced in silico) is important since further confounding issues of the tumour environment are avoided. We demonstrated that the reduced fitness of resistant cells holds both in cell culture and in mouse xenografts. As vascularised tumours are much larger than tumour spheroids, have more complex microenvironmental heterogeneity, and much more limited substrate, we would expect the fitness cost of resistance to be amplified in vivo, thus favouring AT. Further investigation is needed to test this prediction. Clinical outcomes will also depend on the pharmacodynamics of individual drugs, cancer cell type, microenvironment, and mechanisms of resistance[36]. The CDKi that we used is the most specific inhibitor available for CDK2 but does not have sufficient pharmacodynamics to be tested in mice[37].

In tumours, the local microenvironment might contribute to drug resistance, e.g., due to insufficient drug perfusion. Drug gradients in large tumours may be steeper than in our spheroid models, and prolonged exposure to low drug concentrations may facilitate the evolution of intrinsic resistance. Yet the expansion of intrinsically resistant cell populations will always be subject to competition with sensitive cells. Thus, our results are not unduly affected by the presence of environmentally-mediated resistance.

Whereas treatment schedules have been compared in previous computational models of AT[4, 7, 38, 39] and in models incorporating microenvironmental feedback (reviewed in[40]), the relationship between AT and metronomic therapy has received less attention. We thus integrated our general mathematical model of adaptive therapy with an experimentally validated metronomic model accounting for interplay between the tumour and its vascular support[41] (Supplementary Methods). Interesting dynamics arise from this extended model, indicating that AT may be more effective when administered in frequent, low doses than at longer

intervals and higher doses (Supplementary Fig. 9a). This is because the tumour's vascular support recovers, promoting the growth of resistant cells during the breaks in treatment (Supplementary Fig. 9a). When the treatment's antiangiogenic effect is large and the dose frequency is elevated, metronomic therapy with high drug doses compares favourably with AT, but in all other circumstances AT performs best (Supplementary Fig. 9b). Further research is needed to characterise the effects of additional factors, such as immune responses and modulation of resistant cell fitness by environment. For example, whereas our CDKi-resistant cells were more sensitive to hypoxia, this was not the case in erlotinib resistance[35]. Generally, however, AT and metronomic therapy probably both exploit competition between therapy-sensitive and resistant cells as well as effects on the microenvironment.

Our results support the argument that targeting non-essential CDKs that control the cell cycle might be a useful approach for cancer therapy. While specific inhibition of a single CDK is unlikely to be a realistic aim, we have shown that inhibitors with selectivity for CDK2 can effectively limit tumour cell proliferation. We further demonstrated that upregulated CDK6 can compensate for compromised CDK2 functions and the two kinases have similar kinetic parameters. That resistance of cells was maintained for a six-month drug holiday suggests that it was stably encoded in epigenetic modifications. Resistant cells were sensitised to the CDK4/CDK6 inhibitor palbociclib, which is currently approved for therapy of certain breast cancers and, like other CDK4/CDK6 inhibitors, is undergoing clinical trials for other types of cancer[42]. We suggest that, reciprocally, upregulated CDK2 might contribute to palbociclib resistance, a scenario already discovered in acute myeloid leukemia cells with a mutated Flt3 receptor tyrosine kinase[43].

Since CDK1/2 inhibitor-resistant cells are sensitised to CDK4/6 inhibitors, combining both inhibitors could be an advantageous strategy, exploiting a double-bind whereby cells might be unable to evolve resistance to both inhibitors of CDK1/CDK2 and CDK4/CDK6 without drastic reductions in fitness. This need not involve sequential administration of two drugs, as a recent study[44] determined that simultaneous treatment is more effective provided there is no cross-resistance to both drugs. Collateral sensitivity of resistant cells to alternative drugs has recently been validated in experimental models of acute lymphoblastic leukemia and shown by modelling to exploit evolutionary trajectories, much like AT[45]. CDKi-resistant cells were also sensitive to hypoxia and low serum, suggesting other collateral sensitivities that could be exploited by an additional double-bind[38], potentially aiming for cure rather than long-term tumour maintenance. While empirical therapeutic approaches avoid making untested assumptions and will continue to be the mainstay of cancer therapy for the immediate future, mathematical modelling of evolutionary trajectories will take on increasing importance[46].

## Methods

**Cell lines**. The parental and CDK2$^{-/-}$ HCT116 (human colon cancer) cell lines, were purchased from Horizon, UK (HD R02-015). Other cell lines used were: U2OS (human osteosarcoma; purchased from ATCC); BJ-hTERT (human foreskin fibroblasts, immortalized with hTERT; obtained from Dr J. Piette); SK-MEL-28 (human melanoma; gift from Dr Ch. Theillet, IRCM Montpellier). Cells were not authenticated subsequently but were tested for mycoplasma contamination on a weekly basis.

All cells were grown in Dulbecco modified Eagle medium (DMEM–high glucose, pyruvate, GlutaMAX–Gibco® LifeTechnologies) supplemented with 10% fetal bovine serum (SIGMA, HyClone or Pan-Biotech). Cells were grown under standard conditions at 37 °C in a humidified incubator containing 5% CO$_2$. Cell lines were not authenticated in-house but were tested on a weekly basis for mycoplasma contamination.

**Establishing resistant cell lines**. Adaptive-resistant RA10 and RA20 cell lines were obtained by treating HCT116 cells with NU6102 at the initial 2 μM concentration, increased every three days by 2 μM (2>4>6 μM, etc.) until 10 and 20 μM, respectively. R10, R20 and R50 cell lines were grown from the beginning in 10 μM, 20 μM and 50 μM NU6102. Cells were passaged every three days (1/10 dilution) with adding fresh inhibitor.

**Establishing fluorescent cell lines**. HCT116 cells were plated at $1.5 \times 10^4$/cm$^2$ density and transfected 24 h later with eGFP-N1 (1 μg/ml) or pmCherry-N1 (1 μg/ml) vectors, using JetPEI (Polyplus) or lipofectamine 3000 (Invitrogen), respectively, according to the manufacturer's protocol. eGFP-transfected cells were selected with 1 mg/ml G418 for 10 days. GFP and mCherry-expressing cells were sorted with FACS Aria (BD Biosciences, SanJose, CA). Cell sorting was repeated every 6 months due to the loss of GFP expression.

**Drug treatments**. CDK inhibitors were dissolved in DMSO and used at concentrations indicated in figure legends: NU6102 (1–50 μM; Enzo Life Sciences); PD03320991 (0.5–10 μM; Selleckchem); Purvalanol A (0.5–5 μM; Enzo Life Sciences). For analyzing the effects of CDK inhibitors on cell proliferation, cells were plated at density of 150000 or 250000 cells per well in 6-well plates. 6 h later, the medium was replaced with medium containing appropriate concentration of inhibitor.

For AT and MTD/MTDx treatments, mixed cultures of R50-mCh (1%) and WT-GFP + cells were plated at 200000 in 60mm dishes. Cells were treated with either 50 μM NU6102 (MTD: 1 day treatment, 2 days without drug; MTDx: continuous drug treatment), or an initial concentration of 15 μM NU6102 for AT, i.e. just below the concentration required to maintain a stable population. The concentration of NU6102 in AT arm was subsequently modified by ± 20% at 3-day intervals to maintain a constant cell density (70–80%). The proportions of R50-mCh and WT-GFP + were determined by flow cytometry at different time points.

**Cell proliferation assays**. Cells were counted using Muse Cell Analyzer and Muse Cell counting reagent (Millipore) according to manufacturer instructions. Briefly, cells were trypsinised, washed in PBS and resuspended in 1 ml of PBS. Pre-warmed counting reagent (380 μl) was mixed with 20 μl of cell suspension and incubated at RT for 5 min. processing on Cell Analyzer.

**Cell extracts and Western-blotting**. Frozen pellets (harvested by trypsinisation, washed with cold PBS) were lysed with lysis buffer (150 mM NaCl, 50 mM Tris pH 7.5, 0.2% Triton, 1 mM EDTA; freshly added: 1 mM DTT, 0.1 mM NaVO$_4$, protease inhibitors cocktail (Roche)) and incubated on ice for 30 min. Samples were centrifuged for 10 min at 13000 rpm and supernatant collected. Protein concentrations were determined by BCA protein assay (Pierce Biotechnology). Samples were boiled for 5 min in Laemmlli buffer. Equivalent amounts of proteins were separated by SDS–PAGE (usually on 12 cm × 14.5 cm; 7.5% or 12.5% gels). The proteins were semi-dry transferred onto Immobilon membranes (Milipore). Secondary antibodies were either goat antibodies to mouse IgG-HRP (DACO) or donkey antibodies to rabbit IgG-HRP (GE Healthcare). The detection system was Western Lightning Plus-ECL (PerkinElmer) and Amersham Hyperfilm (GE Healthcare). Primary antibodies used were: pRB (G3-245; BD Pharmingen); Rb phospho-S795 (Abcam, ab47474); cyclin A (6E6; Novocastra); cyclin E1 (clone HE12, Santa Cruz Bio.); cyclin D1 (DSC6; Cell Signaling); cyclin D3 (D-7 and B-10; Santa Cruz Bio.); cyclin B1 (GNS1; Santa Cruz Bio.); CDK2 (D-12 and M-2; Santa Cruz Bio.); CDK1 (clone 17; Santa Cruz Bio.); CDK4 (C-22; Santa Cruz Bio.); CDK6 (C-22 and B-10; Santa Cruz Bio.); Cdc6 (180.2; Santa Cruz Bio.); p21 (C-19; Santa Cruz Bio.); Caspase-3 (Cell Signaling); cleaved Caspase-3 (Asp175; Cell Signaling). Uncropped scans are provided in the Supplementary Figs 10–12.

**siRNA transfections**. The SMARTpool: ON-TARGETplus siRNAs (Cdk6, L-003240-00-0005; non-targeting, D-001810-10) were purchased from GE Dharmacon (Lafayette, CO, USA). Cells were transfected with siRNA at 100 nM by calcium phosphate transfection method. Briefly, cells were plated at $1.5 \times 10^4$/cm$^2$ density. 24 h later, medium was changed for medium without antibiotics. Calcium phosphate–DNA coprecipitate was prepared (44 μl H$_2$O, 5 μl 2.5 M CaCl$_2$ and 1 μl 100 μM siRNA). 50 μl CaCl$_2$-siRNA solution was combined with equal volume of 2xHBS buffer (50 mM HEPES, 280 mM NaCl, 1.5 mM Na$_2$HPO$_4$, 10 mM KCl; pH 7,04). Coprecipitates were incubated at room temperature for 1 min, mixed by pipetting, added drop by drop into medium above cells and gently mixed.

**RT-PCR**. Total cellular RNA (1 μg in total volume of 10 μl), extracted by RNeasy Mini Kit (Qiagen), was mixed with 1 μl of 10 mM dNTPs (2.5 mM of each; Life-Technologies) and 1 μl of 50 μM random hexaprimers (New England Biolabs). Samples were incubated at 65 °C for 5 min., then immediately transferred on ice, followed by addition of 5 μl of 5xFirst Strand Buffer, 2 μl 100 mM DTT and 1 μl RNasin® Plus RNase Inhibitor (Promega). Samples were incubated at 25 °C for 10 min. and at 42 °C for 2 min. 1 μl of M-MLV reverse transcriptase (Thermo Fisher Scientific ref. 28025-013) was added to each sample, and incubated at 42 °C for 60 min., then at 70 °C for 15 min.

**qPCR**. qPCR was performed using LightCycler 480 SYBR Green I Master (Roche) and LightCycler 480 qPCR machine. The reaction contained 5 ng cDNA, 2 μl 1 μM qPCR primer pair (final concentration of each primer 200 nM), 5 μl 2x Master Mix, and final volume made up to 10 μl with DNase free water. qPCR was conducted at 95 °C for 10 min, followed by 40 cycles of 95 °C for 20 s, 58 °C for 20 s, and 72 °C for 20 s. The specificity of the reaction was verified by melt curve analysis. Each reaction was performed in three replicates.

qPCR primers (Tm −60 °C): human CDK6 5′-TCAGCTTCTCCGAGGTCT GG-3′, 5′-TAGGTCTTTGCCTAGTTCATCG-3′.

**Flow cytometry - cell cycle analysis**. Cells were harvested, washed once with cold PBS, resuspended in 300 μl cold PBS and fixed with 700 μL chilled 100% ethanol. Cells were kept at −20 °C, at least overnight. On the day of analysis, cells were pelleted by centrifugation at 5000 r.p.m. for 5 min. After washing once with 1% BSA in PBS, cells were stained with Propidium Iodide (PI) solution (10 μg/ml PI, 1% BSA, 200 μg/ml RNase A in PBS) for 30 min. at room temperature, and analysed with BD FACS Calibur (BD Biosciences, SanJose, CA).

**EdU/BrdU incorporation**. To analyse the fraction of replicating cells, cells were either pulse-labeled (15 min.) or incubated for 24 h with either bromodeoxyuridine (BrdU, 200 μM; Sigma Aldrich) or 5-ethynyl-2′-deoxyuridine (EdU, 20 μM; Life-Technologies). Cells were harvested, washed once with cold PBS, resuspended in 300 μl cold PBS and fixed with 700 μl ice-cold 100% ethanol. EdU incorporation was detected using the Click-iT® EdU Alexa Fluor® 488 Imaging Kit (Life-Technologies), and analysed with BD FACS Calibur.

For BrdU detection, cells were washed with cold PBS and permeabilised with 2 N HCl and 0.5% Triton X-100 for 30 min. at room temperature, with occasional vortexing. After adding 5 ml of PBS, cells were pelleted and resuspended in 200 μl anti-BrdU antibody (BD 347580, No.408) at 1:30 in PBS-0.5% Tween 20 and 1% BSA, and incubated 2 h at room temperature. Cells were washed with PBS and incubated with anti-mouse Alexa Fluor 488 (LifeTechnologies) for 2 h at room temperature. After wash in with PBS, cells were resuspended in 500 μl PBS containing 3 μg/ml 7AAD (LifeTechnologies), 200 μg/ml RNAse A (Sigma Aldrich). Samples were incubated for 2 h at room temperature and analysed on FL-1 and FL-3 channel with FACS Calibur.

**Cell competition experiments**. WT and R50 cells (extensively washed prior to the experiment to eliminate the inhibitor) were plated at the indicated ratios (2 million cells in total) in 10cm-dish, without the inhibitor. Cells were harvested every three days, and 1/10 of the mixed cell population was plated again. After harvesting, 1 million cells were washed once with cold PBS, resuspended in 1 ml of PI solution (1% BSA in PBS, 10 μg/ml PI) and analysed on Fortessa flow cytometer (BD Biosciences, SanJose, CA) for the percentage of GFP/mCherry positive cells.

**Microarray analysis–transcriptome**. RNA was prepared from HCT116 WT, CDK2 KO, R50 and WT cells treated with 20 μM NU6102 for 24 h, in duplicates, using RNeasy Mini Kit (Qiagen) following the manufacturer's instructions. RNA was labelled with Cyanin 3 and complementary RNA (cRNA) was synthesized. Cy3-labelled cRNA was amplified and hybridized on the Agilent SurePrint G3 Human GE 8 × 60k Microarray according to the procedures by Hybrigenics Company (Paris, France). Raw data were processed using GeneSpring GX software (Agilent Technologies) to define differently expressed genes, using one-way ANOVA, with a Benjamini-Hochberg corrected p-value < 0.001 and post hoc Student Newman Keuls.

**Human phospho-kinase antibody array**. Human Phospho-Kinase Antibody Array (R&D Systems) is a set of nitrocellulose membranes on which capture and control antibodies for 43 kinases and 2 total proteins have been spotted in duplicates. Cell lysates from WT, Cdk2 KO, R50 and WT cells treated with 20 μM of NU6102 for 24 h, were added to array membranes and processed according to the protocol of R&D Systems. Signal from the membranes was imaged with ECL camera and the intensity of the signal quantified with ImageJ software.

**Live-cell CDK2 activity sensor**. The CDK2 activity sensor was a gift from Sabrina Spencer (Stanford University, CA, USA). The sensor includes 994–1087 amino acids of human DNA helicase B fused to the yellow fluorescent protein mVenus (DHB-Ven) and contains four CDK consensus phosphorylation sites, a nuclear localisation signal and a nuclear export signal[31]. Sensor was transduced into WT, CDK2 KO and R50 cells by lentiviral infection (see below). To obtain stable cell lines, cells were selected for YFP using cell sorter (FACS Aria).

**Lentiviral infection**. Viral particles were produced by transfecting packaging cells HEK293 cells with tat, rev, gag/pol, vsv-g vectors (provided by Dr E. Bertrand, IGMM Montpellier) by calcium phosphate transfection (see above). Cells were plated the day before transfection at density of $4 \times 10^6$ cells in 10 cm plates. Vectors were transfected in the following proportions:

| 20 | : | 1 | : | 1 | : | 1 | : | 2 |
|---|---|---|---|---|---|---|---|---|
| backbone | : | tat | : | rev | : | gag/pol | : | vsv − g |
| 20 μg | | 1 μg | | 1 μg | | 1 μg | | 2 μg = 25 μg total DNA |

The day after transfection, the supernatant from the virus producing cells was recovered, filtered with 0.45 μm filter and centrifuged in 2 ml eppendorfs at 4 °C for 3 h at maximum speed. The supernatant was collected in 50 ml Falcon tube tightly closed on ice in the cold room. The procedure was repeated the following day. For lentiviral infection, HCT-116 WT cells were plated in a 12-well plate, $5 \times 10^4$ cells/well. Cells were rinsed with fresh medium and supplemented with infection mix (300 μl of medium without serum, 6 μg/ml polybrene). Cells were incubated for 2 h at 37 °C, 5% $CO_2$ with occasional tilting (every 20 min). After 2 h, 1 ml of fresh medium was added and cells were left to recover overnight. The next day medium was changed and culture was expanded.

**Immunoprecipitation**. Cell lysates were prepared as described above and 100 μl was used for every immunoprecipitation reaction. Each sample was incubated with 3 μl of appropriate antibody on ice for 2 h, followed by incubation with 50 μl of Sepharose beads (Protein A Sepharose or Protein G Sepharose 4 Fast Flow, GE Healthcare (previously Amersham Biosciences)) on Adams Nutator Mixer at 4 °C for 30 min. Supernatants were collected and saved for analysis. Beads were washed three times with 900 μl of lysis buffer, incubated with 30 μl Laemmli buffer at 37 °C for 15 min., and immunoprecipitated proteins were analysed by Western-blotting.

**In vitro kinase assays**. Wash buffer I: 25 mM Tris pH 7.5, 150 mM NaCl, 0.1% Triton X-100, 1 mM EDTA, 1 mM EGTA, 1 mM DTT, protease inhibitors.

Wash buffer II: 25 mM Tris pH 7.5, 10 mM $MgCl_2$, 1 mM DTT.

Kinase buffer 2x: 100 mM HEPES pH 7.5, 20 mM $MgCl_2$, 2 mM DTT, 0.04% Triton X-100.

Kinase mix 1x (20 μl): 10 μl kinase mix 2×, 2 μl histone H1 (1 mg/ml) (Calbiochem-Merck Millipore) or Rb-CTF peptide (0.678 mg/ml; ProQinase), 1 μl ATP (1 mM), 0.25 μl γATPP[33] (Perkin Elmer), 6.75 μl $H_2O$.

**Kinase assays on immunoprecipitated CDK complexes**. Immunoprecipitations was performed as described above. Beads were washed two times with 500 μl of Wash buffer I and once with the same volume of Wash buffer II. Beads were incubated with kinase mix at 37 °C for 20 min. with occasional tapping the tube. Kinase mix contained either histone H1 (for cyclin A and CDK2) or Rb-CTF peptide (for CDK6/4) as a substrate. Negative controls were kinase mix without kinase (beads only) and IP beads incubated with kinase mix without substrate. The reaction mix was spotted on P81 phosphocellulose (Millipore) paper and washed three times in 1% orthophosphoric acid (10 ml per sample). Papers were air-dried at room temperature and counted with scintillation counter.

**Evaluation of $K_m$ and $K_i$ of recombinant CDK complexes**. Recombinant CDK/cyclin complexes (CDK2/cyclin A2; CDK4/cyclin D1; CDK4/cyclin D3; CDK6/cylin D1; CDK6/cyclin D3) were obtained from ProQinase and used at 6.8ng/μl. Kinase assays were performed in triplicates, with Rb-CTF peptide as a substrate (final concentration 34ng/μl; ProQinase). For measuring $K_m$, assays were performed with ATP at different concentrations (0, 100 M, 200 μM, 500 μM, 1 mM, 2 mM). For $K_i$, kinase assays included NU6102 at 0, 0.2 μM, 0.5 μM, 1 μM, 3 μM, and 10 μM. Assay time was 8 min. Kinetic parameters were calculated using GraphPad Prism software.

**Sensitivity to glucose depletion and hypoxia**. 150000 or 250000 of cells per well were plated in 6-well plates. For measuring the response to hypoxia, the plates were placed in the incubator with 1% $O_2$ (37 °C, 5% $CO_2$) and the number of cells was analysed every 24 h for 3 days. For low glucose and low serum sensitivity analysis, the cells were plated in medium with low glucose (1 g/L), or medium with 1% of FBS, respectively, and the number of cells was counted every 24 h for 3 days.

**Tumour xenografts**. 36 female athymic nude mice (Envigo) of 5 weeks were injected subcutaneously into the right flank with $1.5 \times 10^6$ WT, CDK2 KO or R50 cells, in total volume of 150 μl (12 mice per each cell type). Tumour size and animal weight were measured weekly; mice were sacrificed when tumours reached the size of 1500 mm³. Tumours were dissected and samples were frozen for further protein and DNA analysis. Parts of tumours were embedded into paraffin blocks for immunohistochemistry analysis.

**Multicellular tumour spheroids**. Competitions in 3D were initiated by mixing sensitive (WT) HCT-116 GFP-positive cells with 1% R50-mCherry cells at day 0. 100% WT-GFP and 100% R50-mCherry spheroid cultures were prepared in parallel as controls.

Spheroids were initiated in 96-well plates according to Friedrich et al.[47]. Each well was coated with 50 μl of 1.5% sterile agarose (wt/vol; Sigma, France) in DMEM. Spheroids were initiated by seeding 1500 cells in 200 μl of complete

culture medium (DMEM, 10% FBS) per well. After 96 h, spheroids with a mean diameter of about 350 μm were formed. At day 4, treatment was started by exchanging 50% of the 200 μl medium with 100 μl of fresh medium containing NU6102 (2× final concentration). For control conditions, untreated and DMSO-treated spheroids, 100 μl of media were replaced with fresh culture medium or DMSO-containing fresh medium, respectively. Media were changed every three days as described above. At least 4 spheroids were treated and analysed per condition.

Volume, integrity and fluorescence composition (GFP/mCherry) of each spheroid were monitored starting at day 4 and, every 3 days up to day 28. Phase contrast, GFP and mCherry images were acquired with 2.5x objective using Zeiss Inverted Axiovert 200 M microscope (Carl Zeiss, Germany). All phase contrast images of spheroids were checked and any deformed or irregular shaped spheroids were eliminated from calculations.

Analysis of spheroid volume was performed using ImageJ (v 1.44) with a macro automating size measurements for a folder of phase-contrast spheroid images[48]. The measured area (S) of spheroids 2D projection was used to calculate the radius (R) and the volume (V) of an equivalent sphere.

For flow cytometry, 4–5 spheroids of each condition were collected in 1.5 ml Eppendorf tube and dissociated enzymatically (using 0.05% trypsin, 15 min. treatment), and mechanically (by pipetting up and down three times). The cell suspension was washed with PBS and fixed in 3% paraformaldehyde.

Flow cytometry analysis was performed using a Fortessa flow cytometer (BD Biosciences, SanJose, CA) equipped with blue laser (488 nm) and yellow laser (530 nm). Flow cytometry data were analysed with FlowJo v.10.2 software (LLC 2006–2016). Forward and Side scatter of aggregates of cells were determined using log scale SSC/FSC plots. In general, samples were analysed at a medium flow rate and 10000 events were acquired for each sample.

**Statistical analysis.** Significant differences between experimental groups were determined using an unpaired two-tailed Student $t$-test in Prism 5 (GraphPad). For all analyses, $p$-values < 0.05 were considered statistically significant.

**Calculation of selection coefficients.** Competition between two cell lines can be described using the selection coefficient. If the population sizes are $P$ and $Q$ then the corresponding frequencies are defined as

$$p = \frac{P}{P+Q}, \quad q = \frac{Q}{P+Q} = 1 - p.$$

The selection coefficient of one cell line relative to another is then defined as the rate of change of the ratio of the frequencies. That is,

$$s = \frac{d}{dt} \log \frac{p}{q} = \frac{d}{dt} \log \frac{p}{1-p}. \tag{4}$$

If $s$ is positive then $P$ will increase relative to $Q$; if $s$ is negative then $P$ will decrease relative to $Q$.

If two cell lines in competition grow exponentially and do not interact then the selection coefficient can be predicted from their growth rates[49]. This is because

$$\log \frac{p}{1-p} = \log \frac{P}{Q} = \log \frac{P(0)\exp(r_P t)}{Q(0)\exp(r_Q t)} = \log \frac{P(0)}{Q(0)} + (r_P - r_Q)t,$$

where $r_P$ and $r_Q$ are the growth rates and $P(0)$ and $Q(0)$ are the initial sizes of populations $P$ and $Q$, respectively. Hence

$$s = r_P - r_Q.$$

Accordingly, we predicted the selection coefficient for each competition assay as the difference between the exponential growth rates of the competing cell lines. We then compared this prediction to the selection coefficients calculated from the competition assay frequency dynamics. We made a single prediction whenever growth curves were measured at the same time as competitions were conducted; otherwise we made maximum and minimum predictions based on non-contemporaneous growth curves. We estimated the growth rate of each cell type as the mean slope of log-transformed growth curves during the first 72 h in monolayer culture.

**Adjusting for loss of the GFP marker.** To adjust data for loss of the GFP marker in competitions between GFP + and GFP- subpopulations, we began by normalizing the data (so the two subpopulation sizes summed to unity). We then estimated the rate of loss of the GFP marker by fitting a regression curve to the log-transformed frequency of GFP + cells in the GFP + control assay. We calculated adjustment factors as

$$c(t) = \exp(at + b),$$

where $a$ is the slope of the regression line, $b$ is the intercept, and $t$ is time. We adjusted all GFP+ cell frequencies for loss of the GFP marker by multiplying by $1/c$,

and all GFP- cell frequencies by multiplying by $(1-c)/c$. Finally, we renormalized the data. In no case did this adjustment change a qualitative outcome.

**Non-spatial mathematical model.** Our non-spatial mathematical model of cancer adaptive therapy is inspired by that of Silva et al.[7] However, whereas that paper described population dynamics using recurrence relations, we instead use coupled differential equations, which are easier to parameterize and analyze. We begin with growth equations

$$\frac{dW}{dt} = \lambda_W W, \quad \frac{dR}{dt} = \lambda_R f(R, W) R,$$

where $W$ and $R$ are the chemosensitive and resistant populations, respectively; $\lambda_W$ and $\lambda_R$ are the maximum growth rates; and $f$ is a frequency-dependent relative fitness function. We also consider a model using a Gompertz growth function, which is the most widely-used function for modelling sigmoidal tumour growth curves[50]:

$$\frac{dW}{dt} = \lambda_W W \frac{\log(K/N)}{\log(K/N_0)}, \quad \frac{dR}{dt} = \lambda_R f(R, W) R \frac{\log(K/N)}{\log(K/N_0)},$$

where $N = W + R$ is the total population size, $N_0$ is the initial population size, and $K$ is the carrying capacity.

In our numerical simulations (using the R programming language package deSolve[51]), therapy is applied as a bolus at regular intervals and causes instantaneous cell death. The treatment effect is simulated by multiplying each subpopulation at the time of treatment by

$$\frac{1}{1 + \rho/\text{IC50}_X},$$

where $\rho$ is the dose, and $\text{IC50}_X$ is either $\text{IC50}_W$ (the half maximal inhibitory concentration for sensitive cells) or $\text{IC50}_R$ (the corresponding value for resistant cells). We are interested in the relative benefits of two types of therapy. For maximum tolerated dose (MTD) therapy, every bolus dose is the same. For adaptive therapy (AT) in numerical simulations, the dose is increased (respectively decreased) by 20% if the total population size has increased (respectively decreased) since the previous treatment.

Model analysis and further justification for the choice of frequency-dependent fitness function can be found in Supplementary Methods.

**Non-spatial model with microenvironmental feedback.** To examine how microenvironmental feedback might affect adaptive therapy outcomes, we integrated our frequency-dependent fitness model with an experimentally-validated model of tumour vascularisation developed by Hahnfeldt and colleagues[52]. As before, we assumed that tumour growth is limited by a carrying capacity according to a Gompertz growth function:

$$\frac{dW}{dt} = \lambda_W W \log \frac{K}{N}, \quad \frac{dR}{dt} = \lambda_R f(R, W) R \log \frac{K}{N},$$

where $N = W + R$ is the total population size, and $K$ is the carrying capacity. We further assumed that the carrying capacity is linked to the degree of tumour vascularisation, which can change over time due to an interplay of stimulatory and inhibitory factors:

$$\frac{dK}{dt} = bN - dN^{\frac{2}{3}}K,$$

where $b$ and $d$ represent how strongly the tumour stimulates and inhibits vascularisation, respectively. The second term in the above equation derives from a mathematical analysis of inhibitory factor diffusion from the surface of a three-dimensional tumour[52]. Finally, we assumed that treatment not only kills tumour cells but also inhibits vascularisation. Like in the case of cell death, the treatment effect was simulated by multiplying $K$ at the time of treatment by

$$\frac{1}{1 + \rho/\text{IC50}_K},$$

where $\rho$ is the dose, and $\text{IC50}_K$ is the half maximal inhibitory concentration.

**Spatial computational model.** To simulate the tumour spheroid experimental system, we created a so-called hybrid cellular automaton computational model[53] (written in the C language) in which each cell inhabits a point on a two-dimensional square grid, which represents a cross-section through a three-dimensional tumour spheroid. Cells proliferate and die at rates that depend on local chemical concentrations and cell density.

At the start of each updating loop, all cells that have insufficient oxygen to survive undergo cell death. These dead cells persist (unless replaced by living cells) and form the necrotic core of the tumour spheroid. Next, cells attempt proliferation or (due to effects of the CDK inhibitor) undergo cell death. We use the well-established Gillespie algorithm[54] to select cells and event types, and to determine

the periods between events. According to this algorithm, the probability that cell $k$ will be chosen for either proliferation or death is $(P_k + M_k)/\sum(P_i + M_i)$, where $P_i$ and $M_i$ are proliferation and death rates, respectively. The selection process amounts to sampling with replacement, so the choice of cell is independent of which cells have been selected previously. The chosen cell attempts proliferation with probability $P_k/(P_k + M_k)$, or else undergoes cell death. The time between events is calculated by drawing from an exponential distribution with mean $1/\sum(P_i + M_i)$. For computational efficiency, the diffusion equations are not resolved after every cell proliferation or death event. Instead, events occur sequentially until the number of cells that have undergone division or death reaches 10% of the population size, at which time the diffusion equations are resolved and the proliferation and death rates are recalculated for each cell.

Further details of the computational model are in Supplementary Methods.

**Code availability**. Code used for tumour spheroid simulations is available under a permissive free software license.[32]

**Data availability**. Microarray gene expression data is available on the NCBI GEO database with the accession number GSE102165.

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

## Acknowledgements
This work was supported by Inserm grant No PC201306 and Ligue Nationale Contre le Cancer grant EL2013.LNCC/DF. K.B. was supported by a fellowship from the Ligue Nationale Contre le Cancer. A.S. was supported by a cooperation grant from the French Institute Cairo and Egyptian Science and Technology Development Fund. MH, RN and D.F. were supported by ITMO 'Physique Cancer' (CanEvolve PC201306). Thanks to V. Georget of MRI Montpellier imaging facilities for assistance with light microscopy, and F. Bernex and N. Pirot of the Montpellier Histology Facility (RHEM).

## Author contributions
D.F. and M.E.H. conceived the project. L.K. R.N. and D.F. designed and analysed the experiments and wrote the manuscript. R.N. performed the mathematical modeling. K.B., A.S., O.W.A., B.B. performed the cell biology experiments. S.P. supervised B.B. C.V. performed mouse xenograft experiments.

## Additional information

**Competing interests:** The authors declare no competing financial interests.

