## [Peer Review File · Nature Communications]

Reviewers' comments:

Reviewer #1 (Remarks to the Author):

The authors present an outstanding example of multidisciplinary research that integrates mathematical modeling with empirical studies. This unfortunately remains rare in cancer and should be applauded. Their hypothesis, results, and discussion are excellent and I support publication.

I would suggest that they think about and add some material on the contextual nature of fitness. In particular, the fitness of cells under culture conditions may be very different *in vivo* as the tumor microenvironment characteristically contains limited substrate when compared to the relatively luxurious culture media. As a result, small fitness difference seen *in vitro* can become relatively large *in vivo*.

So, while the experimental work they performed was excellent, I think they should still add a caution that the evolutionary dynamics *in vivo* could be quite different and should ultimately be investigated

Reviewer #2 (Remarks to the Author):

The authors describe a combination of experimental and mathematical research to support the idea that in many situations adaptive therapies can be more effective than conventional MTD ones. To do so they perform *in vitro* experiments using a cyclin-dependent kinase inhibitor (CDKi) as a mean to control the tumour population. Proving that resistance to CDKi is costly in terms of fitness. This is key as AT, and the mathematical model described, rely on this cost to justify its benefit over MTD.

The main conclusion is that the spatial structure of a cancer is what defines the clinical benefit of AT in those circumstances where resistant tumour phenotypes. Counter intuitively the results show that low concentrations of the CDKi can be more effective than high dosages. Finally, this work supports the idea that mathematical modeling of evolution will become increasingly important in cancer treatment and emergence/management of resistance.

There are many commendable elements in this manuscript and I think it combines an interesting approach to explore a therapeutic technique that could be incredibly useful clinically. The main critique I can offer is that the combination of experimental and mathematical models is far from seamless. At this stage is not entirely clear whether the theoretical and experimental models were conceived separately and put together only during manuscript preparation.

From the mathematical modelling perspective the initial non-spatial model struggles with allowing for competitive release of the susceptible population using the idea of frequency dependent fitness to approximate the role of space. This approximation is crude and speculative. It's clear that the authors had known about the importance of space in ATs and wonder whether their initial hypothesis could be been produced using the spatial agent-based model thus presenting the readers with only one mathematical model to understand. Regarding this CA model, while a lot of microenvironmental features could have been considered, the authors went for a sophisticated hybrid CA that models oxygen. O₂ is clearly important and seems to have been modeled carefully but much is speculated about how it is consumed by cells in different active states while seemingly playing a nonexistent role in driving the differences between sensitive and resistant tumor cells and thus, having an unclear impact on the efficacy of the AT.

From the integration of models point of view it feels like more could be explained about how experimental data was used to parameterize the two mathematical models.

From the experimental modelling perspective, the availability of experimental data in the

supplementary section is welcome as it allows for a potential reexamination of the paper's conclusions by other labs.

From the conceptually point of view, both this research as well as the original work by Gatenby and colleagues on which this is based, assume that resistance needs to be costly and is an intrinsic property of the tumour cell but we know understand well the importance of environmentally mediated drug resistance, which might explain how resistant phenotypes emerge even if intrinsic resistance is costly in drug-free environments. Would it be possible for the authors to comment on how this would impact their results?

Also, given the combined experimental/theoretical platform they introduce, I encourage them to explore how metronomic therapies would compare with adaptive ones.

I hope the authors agree with these comments that, hopefully would help the manuscript live to its full potential. As a result I recommend that the authors resubmit the manuscript at a later date which I would be happy to review again.

Reviewers' comments:

Reviewer #1 (Remarks to the Author):

The authors present an outstanding example of multidisciplinary research that integrates mathematical modeling with empirical studies. This unfortunately remains rare in cancer and should be applauded. Their hypothesis, results, and discussion are excellent and I support publication.

We thank the reviewer for his/her appreciation of the manuscript.

I would suggest that they think about and add some material on the contextual nature of fitness. In particular, the fitness of cells under culture conditions may be very different *in vivo* as the tumor microenvironment characteristically contains limited substrate when compared to the relatively luxurious culture media. As a result, small fitness difference seen *in vitro* can become relatively large *in vivo*.

This is an important point, as illustrated by our finding that the R50 cells are sensitised to hypoxia. We did not include this parameter in our modelling of the spheroids, to see how the model behaved more generally, but it is clear that this will have additional impact, in our case making the resistant cells even less fit. This perhaps underlies our experimental results shown in Figure 6 and Supplementary figure 8, and is one of the main points of the manuscript. The fitness differential of resistance is indeed likely to be even more pronounced *in vivo*, a point that we previously mentioned in lines 383-390, although it was probably not explicit enough. We have added to and altered the discussion on this in lines 414-424 and 456-460. That fitness is context-dependent was also found in a recent study on erlotinib resistance, that we have now cited.

So, while the experimental work they performed was excellent, I think they should still add a caution that the evolutionary dynamics *in vivo* could be quite different and should ultimately be investigated

This is good advice and we have added such a caution in lines 425-433 and 456-460.

Reviewer #2 (Remarks to the Author):

The authors describe a combination of experimental and mathematical research to support the idea that in many situations adaptive therapies can be more effective than conventional MTD ones. To do so they perform in vitro experiments using a cyclin-dependent kinase inhibitor (CDKi) as a mean to control the tumour population. Proving that resistance to CDKi is costly in terms of fitness. This is key as AT, and the mathematical model described, rely on this cost to justify its benefit over MTD.

The main conclusion is that the spatial structure of a cancer is what defines the clinical benefit of AT in those circumstances where resistant tumour phenotypes. Counter intuitively the results show that low concentrations of the CDKi can be more effective than high dosages. Finally, this work supports the idea that mathematical modeling of evolution will become increasingly important in cancer treatment and emergence/management of resistance.

There are many commendable elements in this manuscript and I think it combines an interesting approach to explore a therapeutical technique that could be incredibly useful clinically.

We thank the reviewer for his/her appreciation of the manuscript.

The main critique I can offer is that the combination of experimental and mathematical models is far from seamless. At this stage is not entirely clear whether the theoretical and experimental models were conceived separately and put together only during manuscript preparation.

This is a very important point, and we are glad the reviewer brought it up as it made us re-think how to present the results. In fact, the whole project was conceived on the basis of theoretical models, and there was a constant dialogue between the mathematical modelling and experimental results throughout the entire work. Since we originally presented the experimental work prior to the modelling this was undoubtedly difficult to appreciate, leading to the impression that this might have been assembled only during manuscript preparation – which is absolutely not the case! We have now reordered the manuscript in a more logical way, which better reflects the genuine interplay between theory and experiment, in which the modelling precedes the experiments, which are performed to test the models.

From the mathematical modelling perspective the initial non-spatial model struggles with allowing for competitive release of the susceptible population using the idea of frequency

dependent fitness to approximate the role of space. This approximation is crude and speculative.

It's clear that the authors had known about the importance of space in ATs and wonder whether their initial hypothesis could be produced using the spatial agent-based model thus presenting the readers with only one mathematical model to understand.

In restructuring the manuscript we have sought to better communicate the different roles of our mathematical and computational models. We chose to analyse a simple non-spatial model for three reasons. First, a minimally complex model enables us to describe the fundamental dynamics of AT and MTD in general, rather than restricting our analysis to a special case. Second, the model yields approximate mathematical solutions, enabling us to determine in general how treatment outcomes depend on biological parameters. Third, this particular choice of model facilitates comparison between our results and those of a previous study by Silva and colleagues, who used a similar framework. In attaining model simplicity and tractability, we necessarily sacrificed some realism, and we agree that we could have better justified our use of frequency dependent fitness as a proxy for spatial effects. We have therefore added a new section to the supplementary materials and a new supplementary figure (1), justifying in detail our choice of frequency-dependent fitness function. We demonstrate that the particular frequency at which competitive release occurs is not critical to the predictions of the model. We have made complementary changes to the results section.

Whereas our non-spatial mathematical model predicts that the benefit of AT in general depends on the relative fitness of resistant cells when they are rare, our spatial agent-based model enables us to investigate more precisely how such a fitness differential can arise from competition for space and oxygen in a specific biological context. The latter model is more realistic but has much more limited applicability and is more difficult to analyse mathematically. We therefore think that the two models are complementary and that both should be maintained.

Regarding this CA model, while a lot of microenvironmental features could have been considered, the authors went for a sophisticated hybrid CA that models oxygen. O₂ is clearly important and seems to have been modeled carefully but much is speculated about how it is consumed by cells in different active states while seemingly playing a nonexistent role in driving the differences between sensitive and resistant tumor cells and thus, having an unclear impact on the efficacy of the AT.

To perform the modelling we made parsimonious assumptions regarding oxygen diffusion and consumption based on the experimental literature. We purposely avoided including an oxygen-dependent fitness parameter that varies between

sensitive and resistant cells to make the model more general, and we have now inserted text to this effect (lines 322-323). However, our model does take into account oxygen, which, when combined with spatial structure, is an important factor in efficacy of AT, since resistant cells tend to become localised to the interior of the spheroid. We show that when we vary their localisation, AT is compromised (originally Fig. 6f, now Fig. 5f). This should be more clear from the revised version.

From the integration of models point of view it feels like more could be explained about how experimental data was used to parameterize the two mathematical models.

We have expanded our explanation of the model parameterisation in the main text and supplementary methods.

From the experimental modelling perspective, the availability of experimental data in the supplementary section is welcome as it allows for a potential reexamination of the paper's conclusions by other labs.

From the conceptually point of view, both this research as well as the original work by Gatenby and colleagues on which this is based, assume that resistance needs to be costly and is an intrinsic property of the tumour cell but we know understand well the importance of environmentally mediated drug resistance, which might explain how resistant phenotypes emerge even if intrinsic resistance is costly in drug-free environments. Would it be possible for the authors to comment on how this would impact their results?

While many cancer drug resistance mechanisms have been found to be intrinsic to tumour cells, some tumour cells can survive and proliferate if they are present in a compartment where the microenvironment confers protection, e.g. due to low drug perfusion. In this case, the cells remain intrinsically sensitive to the drug. Drug gradients in large tumours may be steeper than those seen in our computational and experimental tumour spheroid models, and prolonged exposure to low drug concentrations may facilitate the evolution of intrinsic resistance. Yet the expansion of intrinsically resistant cell populations will always be subject to competition with sensitive cells. Thus, our results are not unduly affected by the presence of environmentally mediated resistance. We have inserted text to this effect in lines 436-442.

Also, given the combined experimental/theoretical platform they introduce, I encourage them to explore how metronomic therapies would compare with adaptive ones.

This is an excellent suggestion, and we have now done this by integrating our model with a metronomic model where antiangiogenic factors are taken into account. The results (shown in a new Supplementary figure 9) are interesting and show that when the antiangiogenic effect is large and the dose frequency is elevated, metronomic therapy with high doses of drugs compares favourably with AT. In all other circumstances, AT performs best.

I hope the authors agree with these comments that, hopefully would help the manuscript live to its full potential. As a result I recommend that the authors resubmit the manuscript at a later date which I would be happy to review again.

REVIEWERS' COMMENTS:

Reviewer #1 (Remarks to the Author):

The authors have done a very nice job in revising the manuscript. I support publication

Reviewer #2 (Remarks to the Author):

At this stage I consider the manuscript ready for publication. Given the comment in the discussion that combination of CDK1/2 and CDK4/6 could be synergistic I wonder if they have considered the idea that this could be studied as a double bind?

Response to reviewers

REVIEWERS' COMMENTS:

Reviewer #1 (Remarks to the Author):

The authors have done a very nice job in revising the manuscript. I support publication.

Reviewer #2 (Remarks to the Author):

At this stage I consider the manuscript ready for publication. Given the comment in the discussion that combination of CDK1/2 and CDK4/6 could be synergistic I wonder if they have considered the idea that this could be studied as a double bind?

Our response: we thank the reviewers for their constructive reviews. As for the suggestion of combining CDK1/2 and CDK4/6 inhibitors as a double bind, this is indeed what we were thinking and have now made this more explicit in the discussion.